# OpenReview forum: "Contrastive Negative Preference Optimization for Machine Unlearning in LLMs"
_ICLR.cc/2026/Conference — ICLR 2026 Conference Withdrawn Submission_

### Official Review · Reviewer_YKL5 · 2025-10-24

**Soundness:** 3
**Presentation:** 2
**Contribution:** 2
**Rating:** 2
**Confidence:** 2

**Summary:**

This work proposed contrastive negative preference optimization (CNPO) for LLM unlearning. The authors adopted the idea from noisy contrastive estimation (NCE) to derive a novel loss function for unlearning. Mathematically, the proposed loss can be viewed as NPO+a weighted GD-type retain loss, with the weights depending on the similarity between the retain and forget samples. The authors compared CNPO with standard baselines (NPO, SimNPO) on TOFU, MUSE and a new PII benchmark, and showed that CNPO performs the best at balancing between forgetting and utility preservation.

**Strengths:**

I find the idea of contrasting between forget and retain sample for unlearning quite novel and interesting. The authors have conducted extensive experiments to demonstrate the effectiveness of their method. They also curated a new benchmark PII for evaluating unlearning performance, which may benefit future research on LLM unlearning.

**Weaknesses:**

The writing of the paper is less clear. For example, the notation $x$ is used to denote both the prompt and the forget data in some places. In the main text (eq. 4), there are notations like $r(y_r,y_i)$ and $r(x,y)$, and it is unclear to me what the meaning of $x$ is in these places. Please consider using different notations for prompt and response. Also, I cannot find the definition of $d(x,y)$.

**Questions:**

1. The main issue is that in the current eq. 6 (assuming no typo), the softmax weight in the second term does not actually contribute, since when summing over \(i = 1, \dots, k\), the weights sum to one and thus cancel out. In this case, the proposed algorithm is close to NPO + GD retain loss, and there is less novelty.


2. In the PII experiment (Figure 3), what are the baselines used for comparison? Are they the original NPO and SimNPO, or the variants that include a retain loss? If they are the original versions, would incorporating a retain loss lead to better trade-offs?

---

### Official Review · Reviewer_bYNz · 2025-10-25

**Soundness:** 2
**Presentation:** 3
**Contribution:** 2
**Rating:** 2
**Confidence:** 5

**Summary:**

The paper proposes Contrastive Negative Preference Optimization (CNPO), a new preference-based unlearning method for large language models. CNPO integrates contrastive learning into the unlearning objective, jointly considering forget and retain data: retain samples act as positives and forget samples as negatives. Using semantic similarity as a proxy preference signal, CNPO adaptively adjusts unlearning strength.

The method unifies prior approaches, reducing to NPO with many negatives and to GA in the high-temperature limit, ensuring gradient stability. Experiments on TOFU, MUSE, and a PII dataset show that CNPO achieves strong forgetting while better preserving utility compared with GA, NPO, and SimNPO.

**Strengths:**

1. The paper conducts experiments on three unlearning benchmarks, including MUSE, TOFU, and a newly constructed PII dataset, providing a comprehensive evaluation of the proposed method.

2. The mathematical formulations are clear and easy to understand.

**Weaknesses:**

1. The motivation of the paper is insufficient. After introducing the preliminaries of NPO and SimNPO, the authors directly present CNPO without clearly explaining why NCE is needed or how it addresses the limitations of previous methods. An empirical motivation or ablation study demonstrating the necessity of NCE would make the contribution more convincing.

2. The paper mentions SimNPO, which removes the reference model and achieves better performance than NPO. However, CNPO reintroduces the reference model, even though SimNPO shows that it may not be essential. It would be helpful if the authors could clarify why CNPO requires a reference model and whether a reference-free version of CNPO could achieve even better results.

3. Recently, several works have pointed out that LLM unlearning lacks robustness. For instance, relearning attacks using a small portion of the forget set or jailbreaking prompts can easily recover the forgotten knowledge [1]. Moreover, there are emerging robust unlearning methods that combine unlearning with meta-learning [2] or adversarial training [3,4]. It would strengthen the paper if the authors could evaluate the robustness of CNPO under such attacks and compare it with existing robust unlearning approaches.

> [1] Łucki, Jakub, et al. "An adversarial perspective on machine unlearning for ai safety." arXiv preprint arXiv:2409.18025 (2024).
>
> [2] Tamirisa R, Bharathi B, Phan L, et al. Tamper-resistant safeguards for open-weight llms[J]. arXiv preprint arXiv:2408.00761, 2024.
>
> [3] Fan, Chongyu, et al. "Towards llm unlearning resilient to relearning attacks: A sharpness-aware minimization perspective and beyond." arXiv preprint arXiv:2502.05374 (2025).
>
> [4] Sheshadri, Abhay, et al. "Latent adversarial training improves robustness to persistent harmful behaviors in llms." arXiv preprint arXiv:2407.15549 (2024).

**Questions:**

See weakness.

---

### Official Review · Reviewer_829Y · 2025-10-31

**Soundness:** 3
**Presentation:** 3
**Contribution:** 3
**Rating:** 6
**Confidence:** 5

**Summary:**

This paper proposes Contrastive Negative Preference Optimization (CNPO), a new algorithm for machine unlearning in large language models (LLMs). The method aims to balance forgetting target data and retaining general utility by introducing a contrastive preference-based loss that jointly considers relationships between forget and retain samples. The authors derive CNPO from Noisy Contrastive Estimation (NCE) and preference optimization frameworks, showing that it generalizes Gradient Ascent (GA) and Negative Preference Optimization (NPO) under certain limits.

**Strengths:**

1. Clear motivation and theoretical grounding. The paper identifies a real gap in LLM unlearning: existing methods (GA/NPO) overlook structural relationships between forget and retain sets. The proposed CNPO provides a principled derivation based on contrastive preference learning, establishing connections to both NCE and NPO through rigorous theoretical analysis (Theorem 3.1, Proposition 1)

2. The synthetic PII dataset is a valuable contribution to the community, simulating privacy-sensitive unlearning tasks while controlling for linguistic entanglement.

3. Experiments on TOFU, MUSE, and PII datasets show consistent improvements, which demonstrates the effectiveness of proposed method.

**Weaknesses:**

1. While CNPO’s performance is reported, statistical variance and significance are missing.

2. Limited interpretability of metrics. The metrics for fluency, coherence, and PII leakage rely heavily on GPT-4o evaluation, but the setup lacks calibration details (e.g., number of samples, inter-rater consistency).

3. While the paper evaluates CNPO on TOFU, MUSE, and the proposed synthetic PII dataset, it omits key community-standard benchmarks such as WMDP (Wang et al., 2024), which specifically assess malicious-use forgetting and safety retention. The benchmarks included in the paper primarily focus on sequence-level unlearning, where the goal is to remove specific text patterns or samples. In contrast, WMDP emphasizes knowledge-level unlearning, targeting the removal of factual or harmful knowledge while preserving general capabilities. Including such benchmarks would provide a more comprehensive evaluation of CNPO’s effectiveness in real-world unlearning scenarios and demonstrate its robustness beyond pattern-level forgetting.

**Questions:**

Please refer to the Weakness section.

---

### Note · Authors · 2025-12-02

**Comment:**

Dear Program Chair, Area Chairs, and Reviewers,
We would like to withdraw our paper "Contrastive Negative Preference Optimization for Machine Unlearning in LLMs".
We are deeply grateful for the thorough reviews and insightful comments. We agree with the feedback regarding the need for WMDP benchmark and theoretical clarification. We have decided to take the time to substantially revise the manuscript and incorporate these suggestions before submitting it.

Thank you for your time and effort in reviewing our work.

With utmost respect

**Withdrawal Confirmation:**

I have read and agree with the venue's withdrawal policy on behalf of myself and my co-authors.